# Genomic signatures of convergent shifts to plunge-diving behavior in birds

Chad M. Eliason [1,7 ✉], Lauren E. Mellenthin[2], Taylor Hains[1,3,8], Jenna M. McCullough[4], Stacy Pirro[5], Michael J. Andersen[4] & Shannon J. Hackett [6,7]

Understanding the genetic basis of convergence at broad phylogenetic scales remains a key challenge in biology. Kingfishers (Aves: Alcedinidae) are a cosmopolitan avian radiation with diverse colors, diets, and feeding behaviors—including the archetypal plunge-dive into water. Given the sensory and locomotor challenges associated with air-water transitions, kingfishers offer a powerful opportunity to explore the effects of convergent behaviors on the evolution of genomes and phenotypes, as well as direct comparisons between continental and island lineages. Here, we use whole-genome sequencing of 30 diverse kingfisher species to identify the genomic signatures associated with convergent feeding behaviors. We show that species with smaller ranges (i.e., on islands) have experienced stronger demographic fluctuations than those on continents, and that these differences have influenced the dynamics of molecular evolution. Comparative genomic analyses reveal positive selection and genomic convergence in brain and dietary genes in plunge-divers. These findings enhance our understanding of the connections between genotype and phenotype in a diverse avian radiation.

[1] Grainger Bioinformatics Center, The Field Museum, Chicago, IL, USA. [2] Department of Ecology & Evolutionary Biology, Yale University, New Haven, CT, USA. [3] Negaunee Integrative Research Center, The Field Museum, Chicago, IL, USA. [4] Department of Biology and Museum of Southwestern Biology, University of New Mexico, Albuquerque, NM, USA. [5] Iridian Genomes, Inc., 6213 Swords Way, Bethesda, MD, USA. [6] Committee on Evolution Biology, University of Chicago, Chicago, IL, USA. [7] Present address: Negaunee Integrative Research Center, The Field Museum, Chicago, IL, USA. [8] Present address: Committee on Evolution Biology, University of Chicago, Chicago, IL, USA. ✉email: celiason@fieldmuseum.org

Novel behaviors can promote phenotypic diversity by allowing organisms to use resources in new ways, thereby causing selection towards different adaptive peaks[1,2], or hinder phenotypic change through functional constraints[3]. Despite considerable research on morphological changes associated with behavioral convergence[4], less is known about the genomic basis of behavioral convergence[5]. Part of what makes this such a challenge is the complex mapping between genotype and phenotype[6]. Because of this many-to-one mapping, there are many ways that genomic convergence can play out in evolutionary time. For example, we might see convergence in the same amino acid sites (e.g., polar fish evolving antifreeze blood)[7], the same genes but different amino acids (e.g., high altitude phenotypes in hummingbirds)[8], or the same pathways but entirely different genes[9]. Furthermore, the expected number of convergent substitutions across a clade is expected to be low[8] owing to species-specific differences in the genotype-phenotype map[10].

Transitions from air to water pose major sensory and locomotor challenges that affect the evolution of both genes and morphological structures involved in these functions[11]. For example, recent work in aquatic mammals found genomic convergence in response to selective pressures associated with diving and thermoregulation[12,13]. Similarly, loss of function in genes important for taste perception in penguins was suggested to be a consequence of swallowing fish whole[14]. Although many birds eat fish, diving into water at high speeds (i.e., plunge-diving) is rare, evolving in only a few avian families: Pelicanidae (pelicans), Laridae (terns), Phaethontidae (tropicbirds), Sulidae (gannets and boobies), and Alcedinidae (kingfishers). Kingfishers in particular comprise a species-rich clade of birds (116 recognized species[15,16]) with diverse diets–ranging from aquatic to terrestrial prey and including both invertebrates (e.g., crustaceans and insects) and vertebrates (e.g., fishes and squamates)[17]–and multiple evolutionary origins of plunge-diving behavior[18]. Evidence for morphological convergence in beak[18] and brain shape[19] concomitant with these repeated shifts to plunge-diving makes kingfishers an ideal group for studying the genomic basis of convergent phenotypes.

In addition to diverse foraging behaviors, many kingfishers are geographically isolated on oceanic islands recently associated with rapid rates of brain shape evolution[19] and elevated speciation rates[15,20]. Species on islands experience different evolutionary pressures than those on continents[21]. Divergent selection among islands or between the mainland and islands might therefore drive phenotypic changes (e.g., island gigantism) and genomic changes. Alternatively, genetic drift might reduce adaptive potential on islands. For example, genetic diversity of island species is positively correlated with island size[22]. Correspondingly, genetic drift should have an outsized effect on island populations and thus decrease overall genomic convergence. Yet, comparisons of effective population size between continental and island taxa through time, as well as patterns of genomic convergence, have not been thoroughly tested across a species-rich clade. Nearly neutral theory[23] predicts that small populations should experience higher rates of non-synonymous substitutions than large populations[24]. This has been demonstrated empirically in island birds[25], as well as more recently across a broader range of taxa[26]. Because of this, selection is expected to be less efficient in small island populations[27,28].

Here, we use whole-genome data from 30 kingfisher species to assess whether convergent evolution of plunge-diving behavior is associated with convergence at the genomic level. First, we compared temporal patterns of population demographics to test the prediction that island species would have more fluctuations and lower effective population sizes than continental species. Second, we compared genomic convergence associated with a phenotype (i.e., plunge-diving behavior) that has evolved in both continental and island systems to test the prediction that genetic drift associated with small island populations should lessen the strength of genome-wide convergence. Third, we compared levels of positive selection between continental and island plunge-diving species to weigh relative support for drift and selection operating on islands. Given that plunge-diving behavior has emerged independently in both continental and island-dwelling kingfishers, we hypothesized that positive selection would act more strongly on genes related to plunge-diving behavior in continental kingfishers as compared to island species. An alternative hypothesis is that smaller body sizes or environmental heterogeneity among islands[29] might drive increased rates of positive selection on islands. We anticipate our results will have implications for our understanding of convergent evolution across scales and provide a blueprint for further functional genomics work on the sensory capabilities of birds.

## Results

**Effective population size fluctuates more on islands.** Using a large-scale genomic dataset of 30 kingfisher species, including a newly annotated Collared Kingfisher (*Todiramphus chloris collaris*) genome[30], we fit neutral models of sequence evolution in PAML (model M0) to estimate molecular branch lengths for each annotated gene[30]. We then used synonymous mutation rates for each species as input in a pairwise sequentially Markovian coalescent (PSMC) analysis of demographic history through time. Using a recent time-calibrated phylogeny of kingfishers[20], we identified four origins of plunge-diving behavior and five origins of island-dwelling (Fig. 1). Many species living only on oceanic islands had similarly shaped PSMC curves that fluctuated more than continental species (Fig. 1). Rather than focusing on absolute values of effective population sizes (Ne) that are known to be strongly affected by uncertainty in generation times and inaccurate for recent timeframes[31], we instead focused on the overall shape of PSMC curves analyzed in a multivariate phylogenetic comparative framework[32]. These analyses showed no support for the idea that plunge-diving species differ in population size fluctuations through time ($P_{rand} = 0.25$), yet there was significant divergence in curve shape between continents and islands ($P_{rand} < 0.01$; Supplementary Fig. 1), with species living solely on islands showing more drastic decreases in effective population size towards the present (Fig. 1). This result echoes a recent study on population demographics in montane birds in which species with smaller ranges showed greater demographic fluctuations[33]. Convergent evolution of traits that enable species to adapt to unique ecological conditions on islands (e.g., small geographic areas, reduced species diversity, low predation pressures) might be driven by convergent genomic changes, as has been recently documented in boas[34]. However, reduced population sizes on islands might also result in lower convergence in genomes of island taxa. This is because stronger genetic drift in small populations reduces the effectiveness of selection, decreasing the probability of independently evolving the same phenotypic or genetic mutations[35]. Based on this idea, we predicted that island kingfishers would show little evidence for genome-wide convergence.

**Genome-wide convergence is associated with plunge-diving behavior.** To test for genome-wide convergence, we extracted ancestral states of amino acid sequences for each node of the phylogeny from our PAML M0 model fits. For each gene, we then estimated the number of convergent substitutions in plunge-diving lineages (same amino acids evolving from a different ancestral amino acid) and the number of divergent substitutions (i.e., different amino acids evolving from the same ancestral amino acid). Bayesian phylogenetic analyses showed that plunge-diving lineages have significantly more convergent

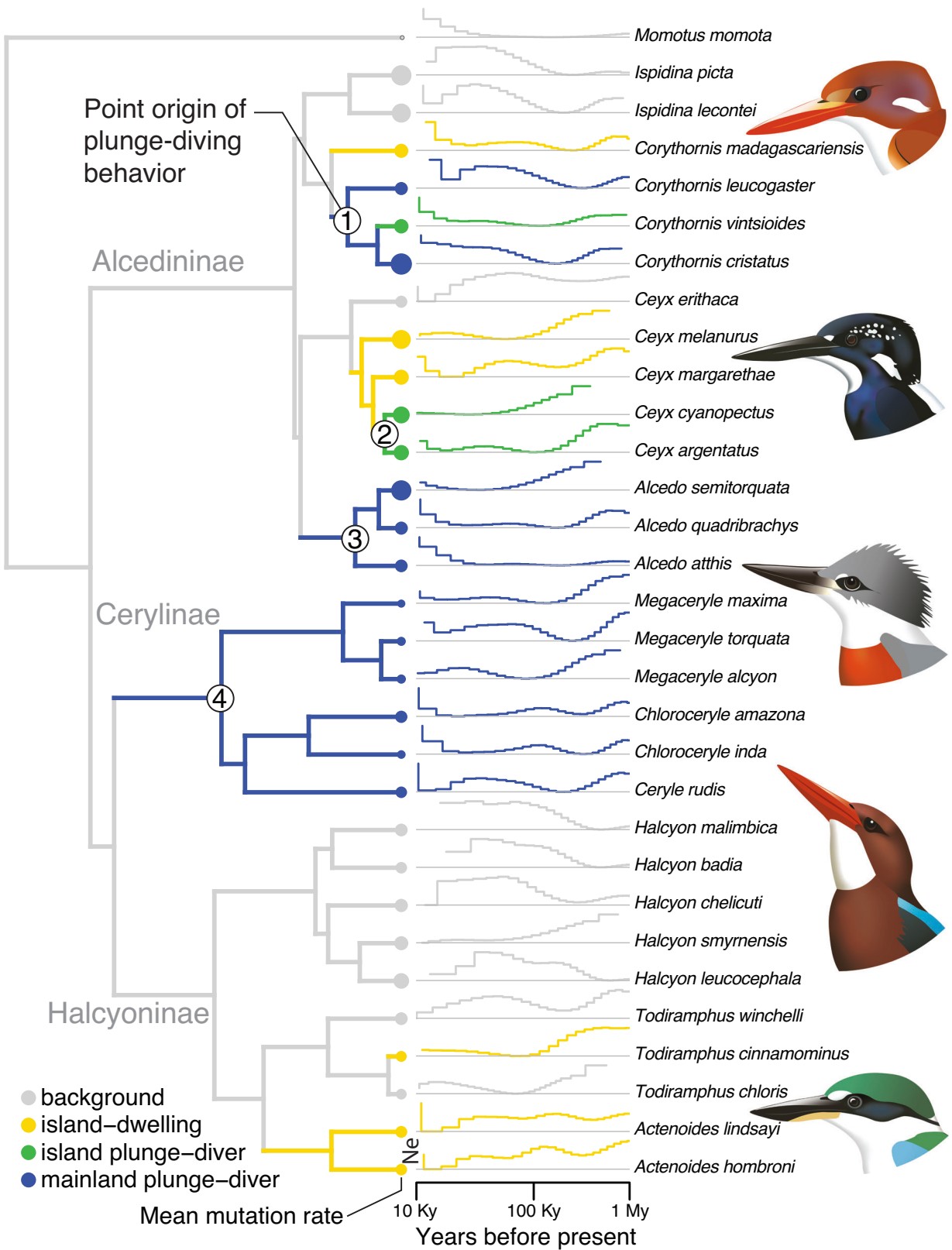

**Fig. 1 Convergent feeding behavior and population demographics of kingfishers.** Phylogeny of kingfisher species sampled with relationships and branch lengths from McCullough et al.[20]. Branch colors correspond to estimated ancestral states of island-dwelling and plunge-diving behavior (see legend). Point sizes at tips correspond to mutation rates estimated from average of PAML M0 trees. Four evolutionary origins of plunge-diving behavior are labeled at nodes. Step plots at right show effective population size (Ne) over time estimated using pairwise sequentially Markovian coalescent (PSMC) modeling. Representative species are (from top to bottom): Madagascar Pygmy Kingfisher (*Corythornis madagascariensis*), Southern Silvery Kingfisher (*Ceyx argentatus*), Ringed Kingfisher (*Megaceryle torquata*), Brown-breasted Kingfisher (*Halcyon gularis*), and Collared Kingfisher (*Todiramphus chloris*). Illustrations created by J. M. M.

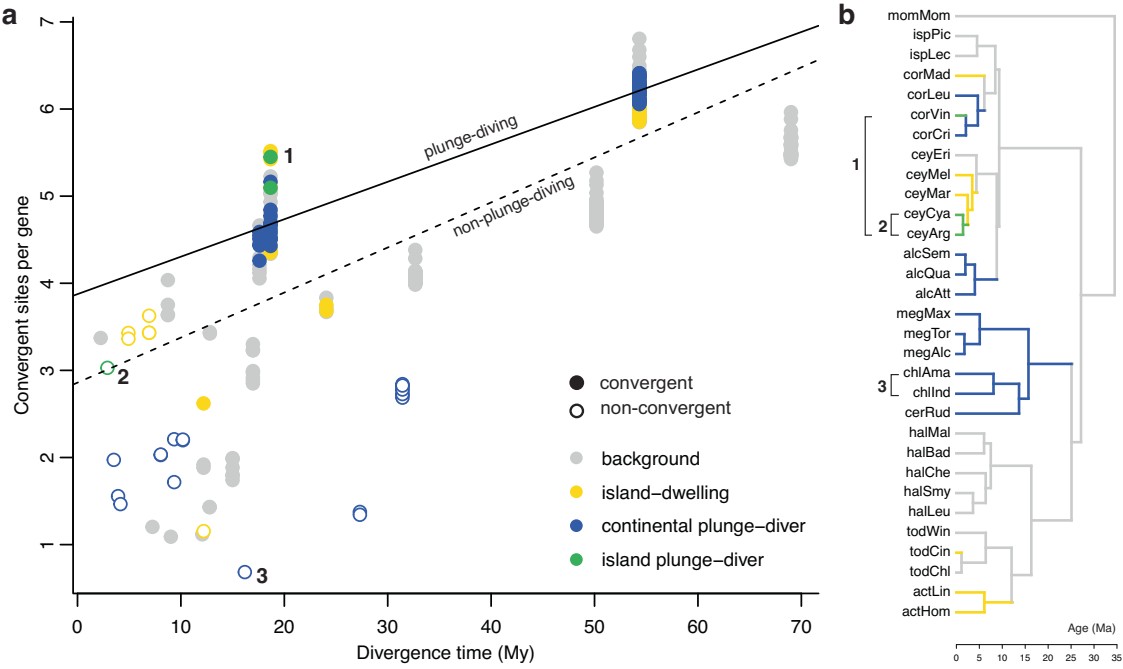

**Fig. 2 Plunge-diving kingfishers are convergent across the genome. a** Scatter plot showing number of convergent sites per gene versus divergence time for all pairs of species. Solid circles denote pairs in the focal set (i.e., those that have convergently evolved plunge-diving behavior or independently colonized islands), and hollow circles denote non-convergent pairs. Point colors correspond to plunge-diving behavior and island-dwelling (see legend). Lines are linear fits to illustrate trends. The effect size for plunge-diving behavior was $\eta^2 = 0.08$, interpreted as a medium effect size[36]. Labels depict exemplar convergent (pair 1: *Corythornis vintsioides* and *Ceyx argentatus*) and non-convergent species pairs (pair 2: *Ceyx cyanopectus* and *Ceyx argentatus*; pair 3: *Chloroceryle Americana* and *Chloroceryle inda*), illustrated on the tree in (**b**).

**Table 1 Predictors of genome-wide convergence in kingfishers.**

| Predictor | Estimate | (95% CI) | $P_{MCMC}$ |
|---|---|---|---|
| Intercept | −1.411 | (−6.354, 3.613) | 0.560 |
| Divergent substitutions | −0.327 | (−0.374, −0.281) | <0.001 |
| Time | 0.112 | (0.062, 0.162) | <0.001 |
| Plunge-diving behavior | 0.062 | (0.012, 0.112) | 0.015 |
| Island | −0.007 | (−0.062, 0.048) | 0.805 |
| Div. substitutions × time | 0.004 | (0.003, 0.005) | <0.001 |

Result of MCMCglmm analysis with phylogenetic relationship incorporated. Response variable is the number of convergent substitutions per gene. Phylogenetic signal of the residuals was 0.81 (95% CI 0.72–0.88). Effect size ($\eta^2$) of plunge-diving behavior was 0.08 (calculated from an ordinary least squares regression fit using lm in R), indicative of a medium-sized effect[36].

amino acid changes than would be expected under a neutral model of sequence evolution in which convergent substitutions scale with divergent substitutions similarly for both plunge-diving and non-plunge-diving lineages (Fig. 2b, Table 1). The effect size ($\eta^2$) for plunge-diving behavior was 0.08 (Table 1), suggestive of a medium effect size[36]. We found no support for genome-wide convergence on islands (Table 1). Given the genome-wide convergence in plunge-divers, we next asked what genes were convergently evolving.

**Several genes are adaptively convergent in plunge-divers.** Identifying genes adaptively converging in plunge-diving lineages requires evidence of both amino acid convergence and positive selection. We looked for signatures of positive selection in plunge-diving lineages using branch-site models[37,38] and protein convergence tests[39] (see Supplementary Fig. 2 for workflow). We identified 6091 convergent genes (Fig. 3b) out of 11938 total genes (Supplementary Fig. 3). Focusing on the smaller subset of genes both convergently evolved and under positive selection in

plunge-diving kingfishers, we identified 93 adaptively convergent genes (Fig. 3b) distributed uniformly across the genome (Fig. 3c). These genes are primarily involved in vasculature development (*ANPEP, CD34, COL3A1, HDAC7, SLC2A10, MIA3*), cognition (*MAPT, AGT, HRH1, DOP1B, SPECC1, SPG11*), chromosome segregation (*BRCA1, BUB1, SMC4, MLH3, NUSAP1, SMAR-CAD1, SPICE1*), retina homeostasis (*ALB, BBS10, NPHP4*), and microtubule-based movement (*MAPT, HYDIN, SPG11, DNAI3, KIF6, NPHP4*) (Fig. 3a; see Supplementary Data 1 for complete gene list).

**Positive selection associated with plunge-diving is stronger on islands.** To test whether island plunge-divers have experienced weaker positive selection than continental plunge-divers, we determined lineage-specific positive selection for each gene and branch on the kingfisher phylogeny (see Supplementary Data 2 for gene list). Multiple list GO term enrichment with metascape[40] revealed island plunge-divers clustered together in GO term co-occurrence space (Fig. 4, Supplementary Data 2). When we analyzed the number of positively selected genes (PSGs), we found that the number of PSGs was significantly related to body size, feeding behavior, and insularity, with a significant interaction between body mass and island-dwelling kingfishers (Fig. 5a, Table 2). Overlap between terms was also significantly higher in island plunge-divers compared to continental plunge-divers (Fig. 5b). Body size is expected to covary with rates of molecular evolution[41], thus the result is not surprising. Given the decreases in effective population sizes on islands (Fig. 1), the elevated positively selected genes in island plunge-diving species are not consistent with nearly neutral theory[24] or recent empirical evidence[27,42], but instead suggest potential shifts in the strength or direction of selection on islands.

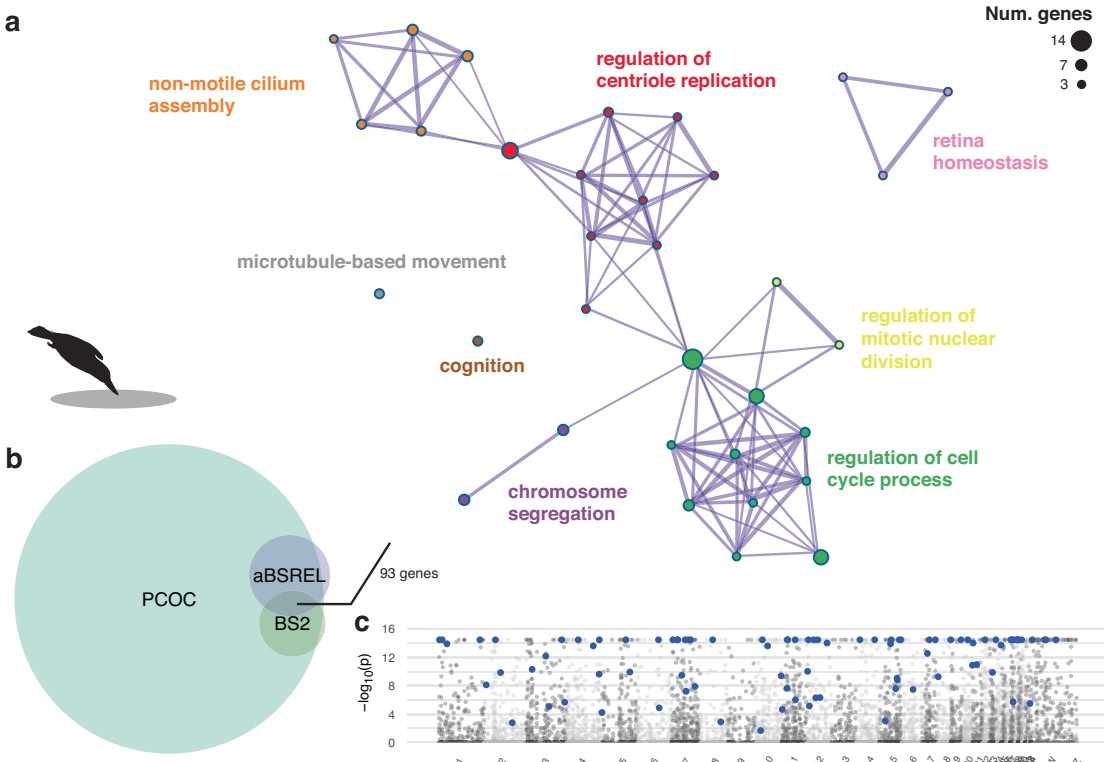

**Fig. 3 Evidence for adaptive genomic convergence in plunge-diving kingfishers.** Euler diagram (**b**) shows overlapping sets of genes that were identified as significant by positive selection tests (BS2, aBSREL) and genomic convergence tests (PCOC). Branch-site (BS2) tests were further filtered to only include those genes also identified as positively selected in plunge-divers under the more stringent Hyphy RELAX model. Size of circles corresponds to the number of genes, with 93 genes identified as significant in all models. **a** Network graph depicting significantly enriched biological functions (FDR < 0.05) in the focal set of 93 adaptively convergent genes. Node size indicates number of genes for that GO term, colors and labels correspond to most significantly enrich GO term for a given cluster. As an example, the microtubule associated protein gene (*MAPT*) that encodes for tau proteins was significant for all four tests (see Supplementary Fig. 11 for sequence details). **c** Manhattan plot showing -log$_{10}$ P values (FDR-corrected) for positively selected genes identified across kingfishers (M2 model; see Supplementary Notes 1, 2, Supplementary Figs. 12, 13, Supplementary Table 2, and Supplementary Data 3 for detailed results), with 93 adaptively convergent genes highlighted in blue (see Supplementary Data 1). Chromosome locations of genes were determined by mapping the scaffold-level collared kingfisher (*Todiramphus chloris collaris*) to a chromosome-level assembly for the Carmine bee-eater (*Merops nubicus*) using liftoff[94]. Photo credit of diving kingfisher: Jean and Fred Hort (CC BY 2.0).

## Discussion

Our results suggest that plunge-diving kingfishers are convergent genome-wide (Fig. 2a) and that specific regions involved in tau protein formation in the brain and other dietary and microtubule genes are driving this pattern (Fig. 3a). This contrasts with recent evidence in anoles[43] and marine mammals[12] that show decoupling between phenotypic and molecular convergence.

Diving into water to forage likely involves adaptations to the eye[44], as well as changes in brain[19] and beak shape[18]. Genetic variation in two retinal genes under positive selection in plunge-divers (*BBS10* and *ATRIP*) have been linked to lower visual acuity in mice[45,46]. Other adaptively convergent genes include *CENPJ*, a gene that has been linked to decreased brain size in primates[47], and *MAPT*, a gene that codes for tau proteins that stabilize microtubules in the brain[48]. Proper functioning of *MAPT* has been suggested to mitigate concussion-related brain injuries during foraging in woodpeckers[49]. This is interesting given that kingfishers show convergent evolution in brain shape that is further tied to their diversification[19], along with evidence of convergence at the genomic level (Fig. 2a). Given the importance of neck musculature and beak shape in species that perform high-speed dives into water[18,50], our findings are consistent with the idea that brain shape and tau protein evolution might be adaptive in allowing plunge-diving species to undergo daily, repetitive concussive forces without sustaining serious injuries. In addition to morphological adaptations, shifts to a fish-eating diet likely involve shifts in dietary physiology[51]. The *AGT* gene in the cognition cluster of enriched genes (Fig. 3a) codes for an enzyme that has recently been linked to dietary variation in other bird species[52], hinting at a potential role in dietary shifts of plunge-diving species. The relationship with chromosome segregation is interesting given some of the highest chromosome numbers have been described in small plunge-diving kingfishers in the Alcedininae subfamily[53]. Further functional genomic work will be needed to test whether these convergent gene changes are associated with adaptive protein function in plunge-diving species.

Our finding of a high proportion of positively selected genes linked to convergent feeding behavior using branch-site tests (Figs. 3a, 5a) echoes work on genes involved with beak shape variation in birds[54], feeding behavior in pandas[55], social behavior in snapping shrimp[56], and extreme environments in cyprinoid fishes[57]. Interestingly, we identified only a small fraction of genes (0.19%) showing support for convergent rate shifts in plunge-diving kingfishers (Supplementary Fig. 4). Similarly low numbers of rate shifts have been found for genes linked to convergent social behavior in spiders[58], bioelectric cues in flying fish[59], and miniaturization in wasps[60]. By contrast, subterranean behavior[61] and hairlessness in mammals[62] have both been associated with high numbers of rapidly evolving genes. This could be due to stronger effects of subterranean behavior on morphology, as compared to plunge-diving behavior. Alternatively, in the case of hairlessness,

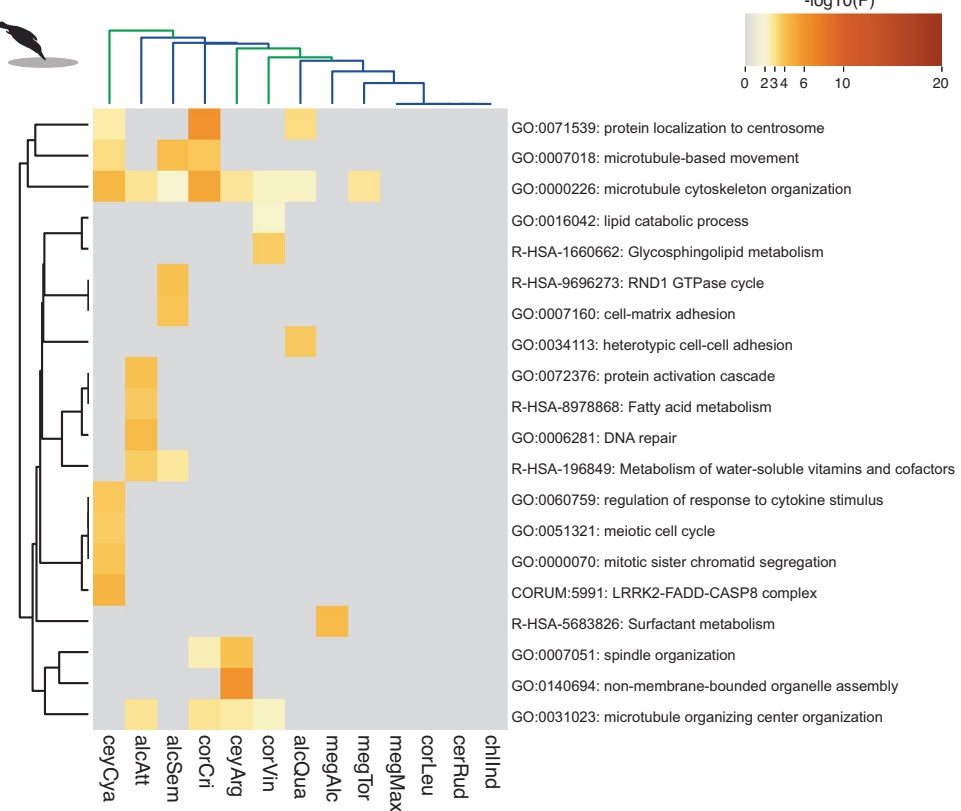

**Fig. 4 Comparing enriched gene functions between continental and island plunge-diving kingfishers.** Clustering analysis of top gene ontology (GO) terms associated with positively selected genes (PSGs) in plunge-diving species as estimated with the HyPhy aBSREL model[83] (see Supplementary Data 2). Note that trees shown are not representative of phylogenetic relationships, but rather depict semantic similarity in GO terms (left) and co-occurrence of positively selected genes associated with these terms (top). Color of cells corresponds to significance level of that GO term enrichment (see legend). Branch colors in upper part of plot correspond to continental (blue) and island-dwelling plunge-divers (green). Plot was produced using metascape[40]. Photo credit of diving kingfisher: Jean and Fred Hort (CC BY 2.0).

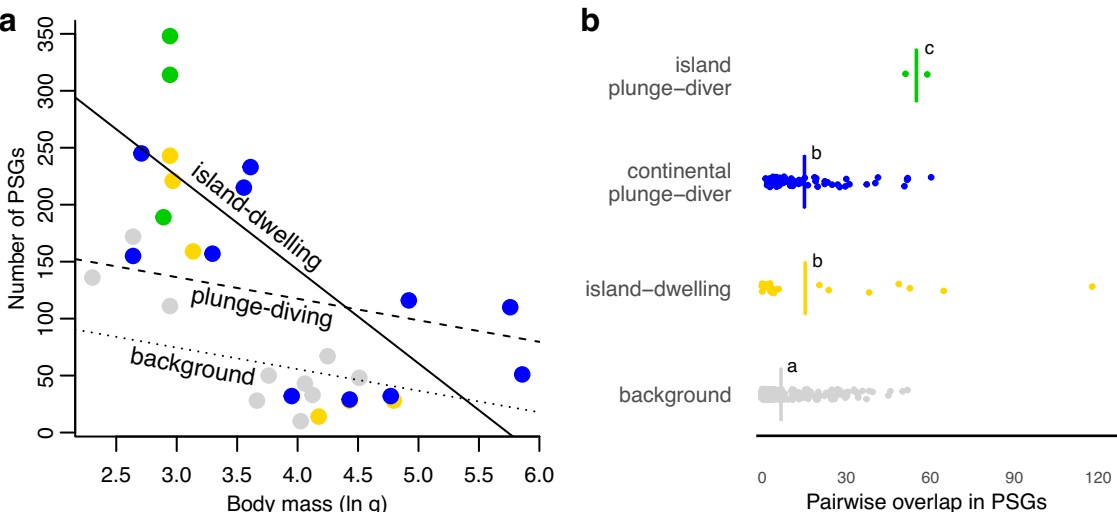

**Fig. 5 Levels of positive selection are highest in small plunge-diving species living on islands. a** Number of positively selected genes (PSGs) versus body size (see legend for point color descriptions). Lines show phylogenetic generalized least squares results of a strongly negative relationship between the number of PSGs and body mass in island kingfishers (solid line) compared to plunge-divers (dashed line; mass-by-insularity interaction, $p = 0.04$) and background (i.e., non-plunge-diving species; dotted line). Note: lines are derived from a PGLS model with moderate phylogenetic signal ($\lambda = 0.55$), thus slopes are shallower than would be expected under an ordinary least squares model without accounting for phylogeny. See Table 2 for statistical details. **b** Overlap in the number of PSGs between pairs of species in the focal (i.e., convergent) and background groups. Points are species pairs and vertical lines are average values. Groups sharing the same letter are not significantly different (i.e., $p > 0.05$).

**Table 2 Predictors of the number of positively selected genes in kingfishers.**

| Predictor | Estimate | Std. err. | t | p |
|---|---|---|---|---|
| Intercept | 131.3 | 73.8 | 1.78 | 0.09 |
| Plunge-diving behavior | 61.8 | 23.0 | 2.68 | 0.01 |
| Body mass (ln g) | −18.9 | 18.2 | −1.04 | 0.31 |
| Insularity | 278.7 | 101.8 | 2.74 | 0.01 |
| Body mass × insularity | −63.3 | 28.9 | −2.19 | 0.04 |

Results of a PGLS model with number of positively selected genes as the response variable. Model $\sigma^2 = 102.6$, phylogenetic signal ($\lambda$) = 0.55, adjusted $R^2 = 0.38$.

morphological changes relate directly to changes in the genome, whereas behavioral traits tend to be more polygenic, meaning they are influenced by multiple genes with smaller individual effect sizes. Consequently, the likelihood of convergent evolution in behavioral traits is reduced, as there are numerous alternative pathways for acquiring the same phenotype. The latter hypothesis predicts a greater number of convergent rate shifts for genes associated with morphology compared to plunge-diving behavior.

Signatures of convergence might also vary depending on what part of the genome is being assessed. Indeed, we found proportionally more accelerated genes than CNEs linked to plunge-diving behavior (see Supplementary Note 3). This differs from recent work on limb loss in reptiles[63], flightlessness in birds[9], and wolves[64] showing a strong effect of convergence in regulatory regions of the genome compared to coding regions. Challenges in comparing genomic convergence across taxa stem from differences in analytical approach (e.g., HyPhy versus PAML, RERconverge versus phyloACC) and in what parts of the genomes are assessed (e.g., coding versus regulatory regions). Overall, studies showing high numbers of rate shifts in genic and regulatory regions tend to be those involving either extreme morphological changes (e.g., loss of limbs or hair)[9,62,63] or behaviors expected to have strong downstream effects on morphology (e.g., loss of flight)[9,65]. Changes in behavior likely involve several trait systems (e.g., sensory, locomotor), which might explain the emerging trend of morphological convergence being linked to stronger signatures of molecular convergence than behavioral convergence. Consistent with past work, we find overlap but also differences in sets of genes identified by different methods (Supplementary Fig. 5). Further work will be needed to test the functions of putative genes identified as critical for feeding adaptations in kingfishers.

Our study provides evidence for adaptive convergence of brain, digestion, and microtubule-related genes associated with convergent plunge-diving feeding behavior and uncovers previously unrecognized diversity of taste receptor genes in kingfishers relative to other avian species. Genomic convergence was linked to plunge-diving behavior but not insularity, consistent with our PSMC results showing reduced population sizes in island taxa that are expected to lower the potential for genome-wide convergence. However, recovered high levels of positive selection in small plunge-diving species suggests that distinct selective pressures on islands could play a role in driving rapid molecular evolution given the clear connection between feeding behavior and island adaptation (e.g., Darwin's finches). Together, these results elucidate genomic bases of behavioral convergence and set the stage for further work on functional genomics and sensory system evolution.

## Methods

**Genome resequencing.** We were loaned tissues from frozen samples at the Field Museum of Natural History (see Supplementary Table 1). We chose species to maximize taxonomic coverage and dietary diversity. We extracted genomic DNA with the Qiagen DNAeasy prep kit according to the manufacturer's directions, and we prepared genomic libraries with the Illumina TruSeq kit using standard adapters. We sequenced libraries at ~70X average coverage (range: 47.7X–95.1X) with 150 bp paired-end reads on an Illumina HiSeq X Ten platform (Genewiz, New Jersey, USA). We removed duplicate reads with SUPERDEDUPER v. 1.3.2 (available at https://github.com/ibest/HTStream). We then used FASTP v. 0.20.1[66] to assess read quality. The majority of reads had low adapter content (<1%) and were of high quality (Q > 30).

**Genome assembly.** We aligned cleaned reads to the draft collared kingfisher genome[30] using BWA-MEM v. 0.7.17-r1188[67]. This read-mapping approach gave a similar result as NextGenMap[68], suitable for highly divergent sequences[69], as well as LAST[70] in a subset of species (Supplementary Fig. 6), therefore we opted for the faster BWA-MEM. We converted alignments to BAM format with SAMtools VIEW v. 1.10 and sorted BAM alignments using SAMtools SORT. We performed INDEL realignment with GATK v. 3.8[71] and finally used BCFTOOLS mpileup v. 1.8 and SAMtools vcfutils.pl vcf2fq to call genotypes and generate individual consensus sequences (i.e., pseudo-genomes). Bases with low coverage (<5X) or low mapping quality (Q < 30) were N-masked in the final consensus sequences. We used the "−I" option in BCFTOOLS mpileup to exclude INDELS because we wanted genomic coordinates to be the same as the reference genome[72]. This allowed us to use the reference genome annotations (GFF format) obtained from GeMoMa and extract coding sequences for each individual. All scripts for data analyses are available at Zenodo[73].

**Reference bias assessment.** To determine the proportion of raw reads mapped to the reference genome, we computed read-mapping statistics with the flagstats function of SAMtools. This showed that between 76.9–99.4% of reads were mapped to the reference genome, with depth of coverage ranging from 11X–20.4X (Supplementary Table 1, Supplementary Fig. 7). Aligning all species reads to the same reference genome can result in mapping bias, in which species-specific reads that are more similar to the reference map more efficiently than those from more distantly related species. To deal with this potential source of bias, we used an N-masking approach[74] in which we N-masked all variable sites and re-mapped reads of three species differing in their phylogenetic divergence to the reference collared kingfisher (Supplementary Fig. 8). If mapping bias is an issue, we would predict that more distantly related species would show a significant increase in the proportion of mapped reads after N-masking, whereas closely related species would not be affected by N-masking. Our results did not show this pattern, suggesting that reference bias is not strongly influencing our results (Supplementary Figs. 8, 9).

**Population demographic history.** To understand population demographic fluctuations over time, we used PSMC v. 0.6.5[75]. We used PSMC settings (-N25 -t15 -r5 -p'4 + 25 * 2 + 4 + 6') from a recent study on penguin genomes that used a similar assembly approach to ours[72]. To plot effective population sizes (Ne) as a function of time, we used published R code[76] to input data into R with a generation time of 4.8 years (IUCN 2021) and mutation rates taken from the average substitution rate estimated with the PAML M0 model (see below). These rates were comparable to the value used for the common kingfisher (*Alcedo atthis*, 0.11 mutations per million years) in a larger study on the evolution of mutation rates[77]. To statistically test whether the shapes of PSMC curves differ between islands and continents, we used a multivariate clustering approach. Briefly, we first determined the time range with Ne data for all 31 species. Next, we log-transformed the Ne values. Finally, after converting these data into an array,

we used the procD.pgls function[32] in the R package geomorph[78] to run a distance-based phylogenetic generalized least squares analysis, with either insularity or feeding behavior as a predictor and a recently published kingfisher timetree[20].

**Assessing genome-wide convergence**. We tested the hypothesis that plunge-diving behavior results in genome-wide convergence following Sackton et al.[9]. Briefly, we determined the total number of convergent amino acids (i.e., changes in specific amino acids having evolved convergently in two species compared to the most recent common ancestor, MRCA) and divergent amino acids (i.e., amino acids that differ in two species as compared to their MRCA) for each pair of lineages. In calculating convergent and divergent sites, we opted to treat lineages as the whole path from an ancestor to tip rather than single branches[79]. We then used phylogenetic linear mixed models implemented in MCMCglmm[80] to test whether the number of convergent changes was explained by feeding behavior, island dwelling, divergence time, number of divergent amino acid substitutions, or the interaction between divergent substitutions and divergence time. Given potential inflation of effect sizes due to non-independent samples (e.g., single lineage compared to multiple other lineages), we treated each lineage in a pairwise comparison as a random effect in the model, after shuffling lineage pairs to ensure even occurrence of each lineage in a comparison[81]. We also accounted for phylogenetic non-independence by treating the MRCA of each pair as a random effect, with an expected covariance structure determined by the phylogenetic tree calculated with the ginverse function in MCMCglmm. We ran the model for $10^6$ generations and discarded the first 25% as burn-in. Phylogenetic signal was calculated as the posterior mean of the node random effect divided by the sum of all random effects scaled by $(1/\pi^{2/3})$[81].

To further test which genes were convergently evolved in plunge-diving kingfishers, we used the program PCOC[39]. This model tests for signatures of convergence in all residues of a protein. We set up the cutoff of the posterior probability that a given residue is evolving convergently in plunge-diving lineages to 0.8. Genes that showed support for convergence were retained and used in downstream enrichment analyses (see below).

**Detecting signatures of positive selection**. Positive selection in DNA sequences is typically tested by estimating $\omega$, the ratio of the rate of nonsynonymous ($d_N$) nucleotide substitutions to that of synonymous substitutions ($d_S$) for aligned homologous protein sequences[37]. Support for $\omega > 1$ is interpreted as evidence for positive selection, whereas $\omega < 1$ suggests purifying selection (i.e., selective constraints) and $\omega = 1$ indicates neutral evolution. For each aligned coding sequence, we used PAML[37] to fit a null model (model M0) of sequence evolution assuming a single $\omega$ across the genome and phylogeny. For this model, we used a fixed species tree previously estimated using a large UCE data set[20]. To test whether some genes show a pattern consistent with positive selection, we compared two sets of site models that allow $\omega$ to vary across sites in a protein but not across the phylogeny. The first set of comparisons was between a nearly neutral model that assumes two site categories ($\omega 0 < 1$, $\omega 1 = 1$; model M1a) and a positive selection model that allows three site categories ($\omega 0 < 1$, $\omega 1 = 1$, $\omega 2 > 1$; model M2a). The second comparison of site models was between a model that allows ten site categories with $\omega < 1$ along a beta distribution (model M7) and a model that allows 11 site categories, including one with $\omega > 1$ (model M8). Finally, to test the hypothesis that some genes and sites show positive selection in species that plunge dive into water relative to species that do not, we fit a branch-site model that allows for $\omega > 1$ only in predefined foreground (i.e., plunge-diving) lineages (model BS2) to one that assumes $\omega = 1$ for both plunge-diving and

non-plunge-diving species (model BS1). For all genes and models, we used the molecular branch lengths estimated under the neutral model M0 to speed up computational time[9]. We compared sets of models using likelihood ratio tests and retained ancestral state estimates under the M0 model for convergent evolution tests (see below). Models were run iteratively over all annotated genes using custom R scripts[73]. To account for multiple comparisons, we adjusted P values using false discovery rates (FDR) and retained all genes with significant evidence (FDR < 0.05) for positive selection. Sets of positively selected genes for each model comparison were retained and used in subsequent gene enrichment tests.

We are aware of recent work suggesting that branch-site models can overestimate the number of significantly converged genes[82]. This is because branch-site models do not account for potential background positive selection independent of the focal lineages (i.e., plunge-divers). To account for this effect, we further fit Hyphy RELAX models[38] with the same trees used in our branch-site tests. This model is more stringent in testing whether convergent shifts in amino acid dN/dS ratios are sensitive to background lineages. We then filtered branch-site results to include only those genes that also showed significant support for positive selection in foreground lineages under the RELAX model. In total, we confirmed that 308 out of 1426 genes (21.6%) that showed support for positive selection in plunge-divers under the branch-site model were also significant in the more stringent RELAX model.

We also fit the adaptive branch-site random effects likelihood (aBSREL) model[83] in HyPhy. This test conducts significance tests for each branch and gene. To compare the set of positively selected genes associated with plunge-diving behavior among CODEML models and HyPhy models, we retained gene-specific aBSREL tests that showed any significant shifts along a branch (Supplementary Fig. 10). For the aBSREL results, we filtered positively selected genes to include those that were convergently positively selected in at least two lineages (e.g., *Ceyx argentatus* and *Corythornis vintsioides*; see Fig. 1). This resulted in a set of 418 genes showing support for convergent positive selection. Finally, to determine the set of genes that were adaptively convergent in plunge-divers, we required that a gene be present in the PCOC, RELAX, aBSREL convergent, and branch-site (BS2) gene sets. This resulted in a final set of 93 putative adaptively convergent genes (Fig. 3b; see Supplementary Fig. 2 for further details).

**Alternative approaches for identifying convergent positively selected genes**. We used several approaches to identify sets of genes putatively convergent and positively selected in plunge-diving kingfishers. In addition to the previous analyses, we also used (i) a modified branch-site test that accounts for the issue of positive selection in background species in the codeml BS2 model and (ii) a recent method for detecting Combinatorial SUBSTitutions of codon sequences (CSUBST) that identifies genes in pairs of species that are both convergent and under positive selection[84]. For the latter, we filtered genes to only include those that showed support for adaptive convergence in two or more pairs of species that have convergently evolved plunge-diving. See Supplementary Note 4 and Supplementary Data 4, 5 for results.

**Comparing positive selection between islands and continents**. We tabulated the number of positively selected genes for each plunge-diving species branch from the aBSREL results and then used metascape[40] to analyze the co-occurrence of different gene functions among species. To test whether positive selection is operating more strongly in island-dwelling kingfishers, we calculated the number of PSGs for each species as well as the amount of overlap between pairs of species. Since small body sizes are associated with elevated rates of molecular evolution[41], we included

body mass (ln grams) as a covariate in the model, along with insularity, plunge-diving behavior, and their interactions with body size. We determined the best-fitting model using bi-directional variable removal based on AIC scores in the phylostep function[85]. For gene overlap, we used Bayesian mixed models implemented in MCMCglmm[80] with number of overlapping PSGs as the response; insularity and plunge-diving behavior as predictors; and species 1, species 2, and the node of the most recent common ancestor in the phylogeny as random effects (see ref. [81] for details).

**Gene enrichment**. To understand whether sets of positively selected or convergent genes were associated with certain molecular functions, we used metascape[40] with default settings and a custom background of collared kingfisher genes. This analysis links genes with their putative functions (i.e., gene ontology, or GO terms) and tests whether a set of target genes with a given function is larger than expected by chance from a random subset of GO terms. We visualized overlapping target genes from different analyses using Venn diagrams and visualized gene networks again with metascape[40]. Significance of enriched gene functions was estimated using the FDR metric.

**Identifying conserved noncoding elements**. Recent work has found that regulatory regions of the genome, rather than coding regions, are related to shifts in phenotype[9,54]. To assess whether regulatory regions evolve faster in plunge-diving or island-dwelling kingfishers, we identified a set of avian-specific highly conserved noncoding elements (CNEs)[86]. Since these elements are relative to the chicken genome, we used LAST[70] to align the draft collared kingfisher genome[30] to the chicken genome. We then used liftOver[87] to convert chicken genome coordinates to collared kingfisher coordinates and extracted DNA sequences with bedtools getfasta[88]. In total, we located 235559 out of 265983 (88.6%) chicken CNEs in the collared kingfisher genome. After filtering out CNEs smaller than 50 bp and removing badly aligned regions with trimAL[89], we fit estimated molecular branch lengths using the BASEML free model[37]. This model lets each branch assume its own rate of molecular evolution, a requirement of downstream RERconverge analyses. In total, we obtained a final set of 39618 trees. For those CNEs that showed significant support for a convergent rate shift in focal lineages, we used bedtools closest to identify the closest annotated gene in the collared kingfisher genome and input these annotations into GO term enrichment analyses.

**Detecting convergent shifts in rates of molecular evolution**. To detect significant accelerations or decelerations associated with plunge-diving behavior, we ran RERconverge[90] on sets of gene and CNE trees. Importantly, these analyses utilize overall rates of molecular evolution and do not account for variation in specific sites along a gene or CNE. To assess significance of relationships between rates of molecular evolution and phenotypic changes, we used a phylogenetic permulation approach recently shown to be more rigorous than $P$ values derived from parametric estimates[91]. This method simulates evolutionary histories of a given phenotype (e.g., plunge-diving behavior) and then generates a null distribution of correlation values between the phenotype and rates of molecular evolution. Statistical significance is then assessed by determining the proportion of null values that are less extreme than the empirical correlation for a given gene. We did 500 permulations and corrected $P$ values using FDR.

**Statistics and reproducibility**. Maximum likelihood approaches were used to determine support for significant positively selected genes, with correction for multiple tests using FDR ($n = 11938$ genes tested). Bayesian phylogenetic linear mixed models were used

to analyze genomic convergence and gene overlap ($n = 465$ species pairs). Phylogenetic generalized least squares analysis was used to analyze positively selected gene counts ($n = 30$ species). All statistical analyses were performed in R v. 4.2.3. Code was written in the R markdown format to allow for reproducibility of all analyses and generation of figures from raw data sets.

**Reporting summary**. Further information on research design is available in the Nature Portfolio Reporting Summary linked to this article.

## Data availability
Raw Illumina reads are available publicly at the Sequence Read Archive (see Supplementary Table 1 for SRR IDs). Ecological and source data underlying figures are available on Dryad (https://doi.org/10.5061/dryad.gf1vhhmvn)[92]. All genome alignments are available on Zenodo as BAM files (https://doi.org/10.5281/zenodo.7872534)[93].

## Code availability
All code needed to run analyses is available on Zenodo (https://doi.org/10.5281/zenodo.8291004)[73].

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

## Acknowledgements

We thank Isabel Distefano for helping with DNA preps. Funding for genome sequencing was provided by Iridian Genomes, grant# IRGEN_RG_2021-1345 Genomic Studies of Eukaryotic Taxa. This work was partially supported by the National Science Foundation (NSF EP 2112468 to C.M.E. and S.J.H., NSF EP 2112467 and DEB 1557051 to M.J.A.).

## Author contributions

C.M.E. designed research; C.M.E., L.E.M., T.H. and S.P. performed research; C.M.E., L.E.M. and T.H. analyzed data; C.M.E., L.E.M., T.H., J.M.M., S.P., M.J.A. and S.J.H. wrote the initial manuscript; C.M.E., J.M.M., T.H. and M.J.A. revised the manuscript.

## Competing interests

The authors declare no competing interests.
