## [Peer Review File · Communications Biology]

Reviewers' comments:

Reviewer #1 (Remarks to the Author):

In this paper, the authors use a large set of 31 whole genomes of different species of kingfishers that independently evolved plunge-diving behaviors on islands or continental locations (representing populations of these species with different types of demographic history). The authors found that different types of kingfishers showed evidence of different overall demographic histories on islands and continents, and correspondingly, island plunge-divers had stronger signatures of positive selection than continental species. They found several different types of genes associated with convergence in plunge-diving birds, and that there is the rapid evolution of taste receptors, but no evidence that these shifts are associated with tongue shape.

Major Concerns:

Overall the authors did a thorough job of analyses, and use robust combinations of methods that support their findings. I also applaud them on a streamlined manuscript that is clearly written. My biggest concerns with this paper are not about the methods that they used, but about having had enough detail on the methodology to be able to figure out in detail what they did. Although I appreciate that it is the format of the paper, it was difficult to read the results section alone and understand what was going on without going back and reading the corresponding methods section. So, overall, I would recommend that the authors present a bit more of the methods when presenting the results. There was one section that especially needed more information on the methodology in the paper overall to be able to understand, which I go into further below:

Methods Questions in the "Effective population size fluctuates more on islands" section:

- Was the topology of the phylogeny of these species estimated separately and held constant for the M0 model? If so, how was the species tree of those relationships estimated?
- The comparative analyses of the PSMC results were compelling, but I didn't totally understand how they were conducted with the methodological detail that was provided. For example, how did they score the shape of the curve? For the number of population fluctuations, how was that determined? When comparing curve shape, did they consider uncertainty?
- Did the authors estimate effective population sizes of mainland versus island taxa, or was that inferred from the PSMC results? PSMC is not a good program for estimating modern population sizes, so they should be calculated separately (they could compare levels of heterozygosity among island and non-island taxa). Looking at Figure 1, it appears that in several cases there are big jumps to higher N_e , for example, in *Actenoides lindsayi*.

Minor Concerns:

Introduction, 4th paragraph: I would recommend reordering this paragraph, so the impacts of being on an island on patterns of genetic diversity and genomic convergence are closer to that introduction, before jumping in on speciation rates. Similarly, in the last sentence of that paragraph, do you mean genes related to plunge diving?

Results, "Several genes are adaptively convergent in plunge-divers" section: How many genes were tested, and how many genes were identified as being under positive selection? I also think that the second paragraph describing specific functions would be more suited to the discussion versus the Results section.

Results, "Positive selection associated with plunge-diving is stronger on islands" – When talking about how island plunge-divers cluster together, is that relative to all species, or to other plunge-diverse? I think it would help the reader by mentioning that the tree shown isn't representative of genetic

relationships, but rather similar go-terms.

Results, "Convergence in tongue shape is largely decoupled from taste receptor evolution": The reader could benefit from a brief description of the hyoid bone and how it might relate to important tongue morphology.

Figure 2 – the tree diagram showing convergent versus non-convergent pairs is very helpful, but it would be helpful to also define them briefly with words.

Reviewer #2 (Remarks to the Author):

The manuscript entitled "Genomic signatures of convergent shifts in feeding behavior are stronger on islands" by Eliason et al. explored patterns of convergent evolution associated with the feeding behavior of 31 species of kingfishers. The authors report significant levels of convergent evolution, especially for species with plunge-dive behavior. The authors also report that island species had more dynamic effective population sizes through time and maintained smaller populations, which can directly impact patterns of positive selection. The genes under selection in the targeted species were associated with GO terms potentially important for the evolution of their behaviors. In addition to the genome-wide scans, the author found signals of positive selection in genes associated with taste receptors linked to fish-eating behavior in other species of birds. The manuscript is quite interesting, given that it explores the genomic architecture behind distinct feeding behaviors, which is not that common in the literature. Although the associations they found are not proof of the function of the genes, the authors have paved the way for future functional studies targeting specific genes and pathways. The manuscript is well-written, and the main ideas being tested are clear and easy to understand. Most of my comments are minor. For instance, I missed more detailed explanations on particular subjects that might help readers understand the main components of the paper (see comments below). The methods used in the manuscript are adequate, however, I was not able to check the reproducibility of the study given that the Dryad link with the scripts was not available. I found the reduced association between CNEs and targeted traits quite interesting. On the evolutionary scale of the study, regulatory elements should play an important role in determining targeted traits. I wonder if there is an alternative way to map CNEs to the species' genomes other than using blast hits on the chicken genome. Maybe select CNEs between species that are more closely related to the targeted group?

Minor comments:

L109: Explain how it might influence convergence. Lower N_e makes selection less effective, and we might expect reduced convergent patterns.

L116: I think it is important to mention nearly neutral theory (Ohta 1973) here.

L122: I think it is important to mention here the other variables associated with island species that can impact evolutionary rates (e.g., smaller sizes).

L139: It would be good to describe in the text (not only in Fig. 1) how many instances of plunge-diving behavior evolved in the system and how many times it occurred on islands vs. continents.

L158: I think it is essential to include here the fact that drift is stronger in smaller populations, reducing the effectiveness of selection.

L163: An interesting piece of information that could easily be added to the paper is the location of genes associated with convergent evolution. Are there clusters of genes, or are they widespread across the genome? I wonder if genes with convergent amino acid substitutions in plunge-diving species are "linked" or not.

L226: How does this relate to nearly neutral theory and mutation load?

L399: Can you provide a graph showing the relationship between % of mapped reads and divergence time from the reference?

Figure S3: I cannot read this figure. The information is too small.

Reviewer #3 (Remarks to the Author):

The article "Genomic signatures of convergent shifts in feeding behavior are stronger on islands" by Eliason et al. investigates signatures of genomic and morphological convergence in plunge-dive behavior across kingfishers and attempts to correlate these signatures with biogeographic characteristics of the species, specifically differentiating between continental and island-dwelling species. While the study offers an intriguing and relevant question, a robust dataset comprising 31 kingfisher genomes accompanied by morphometric measurements, and thorough analyses, there are certain concerns regarding the manuscript's focus, narrative, and hypotheses that need to be addressed. In the following sections, I will provide a detailed account of these concerns.

One of the primary weaknesses of the manuscript lies in its lack of focus. Although the authors have conducted a series of intriguing analyses, such as identifying amino acid changes occurring convergently across plunge-diver species, detecting signatures of convergent positive selection, testing the frequency of these genetic convergences in islands versus continents, examining signatures of positive selection in taste receptor genes, and exploring the correlation between tongue shape and evolutionary rates in taste receptor genes, these analyses seem disjointed and lacking a cohesive narrative. The introduction, in particular, fails to effectively establish how these questions will be addressed later in the text, giving the impression that the various analyses were just randomly assembled into a single manuscript. The introduction would benefit tremendously from a major rewriting that not only establishes the research questions but also provides a stronger background to justify the hypotheses being tested. For example, why would living on islands and having lower effective population size influence the likelihood of observing genomic convergence? Why does taste receptor gene evolution and its correlation with tongue morphology matter for the big picture of the article?

Another major concern pertains to the notion that species living on islands show smaller effective population sizes and therefore more positive selection and overall genomic convergence. My understanding is that, according to neutral theory, a lower effective population size implies that natural selection is less effective due to the predominant influence of genetic drift. Genetic drift can lead to the removal of adaptive variants or an increase in the frequency of deleterious alleles. This concept is distinct from the strength of selection, which is determined by environmental pressures (either biotic or abiotic) and can, in fact, be stronger on islands compared to continents (due to, for instance, intense competition for limited resources or other island's ecological dynamics). Based on these considerations, it is reasonable to expect that a lower long-term effective population size on islands would result in fewer signatures of selection, unless the strength of selection is exceptionally high. On top of that, I am struggling to understand the rationale behind the idea that genomic convergence should occur more often on islands. While it may be logical to expect increased convergence if island species exhibit a higher propensity for hybridization or if they share a common ancestor, I can't see why being on an island could influence the chances of evolving a trait through the same genetic change. The article would improve substantially if these points were clarified.

My third concern is with the recurrent statement suggesting that a behavior or trait can "cause" molecular convergence (see L81, L154, L414). I think the reasoning here is completely inverted. Certain traits and behaviors can evolve due to convergent molecular changes and not the other way around. Please revise the text to accurately reflect the relationship between molecular convergence and the emergence of traits or behaviors, emphasizing that convergent molecular changes are the underlying drivers that enable the evolution of similar traits or behaviors across different species. Another major point concerns the analysis of convergence associated with plunge-diving behavior. As suggested in Foote et al. (2015), cited in the manuscript, both neutral and adaptive substitutions occurring in genes are limited because of constraints related to deleterious or pleiotropic effects of these mutations. Consequently, coding sequence convergence can occur even in the absence of apparent shared positive selection. However, methods such as codeml in PAML, which are commonly

employed to detect rates of change, are unable to test the null hypothesis that the foreground species (in this case, the plunge-diving species) have no relationship with the observed convergence in coding sequence. To adequately test the hypothesis of convergence, a recommended approach would be to employ a permutation test where the foreground species would be randomized, and the analysis would be repeated multiple times. Then you could use the distribution of the number of convergent changes to infer whether the empirical result falls beyond what is expected by chance.

Finally, I would like to raise the issue concerning the small sample size of plunge-diving species occurring on islands, which consists of only three species. Beyond the statistical limitation, I have reservations about the robustness of the results, particularly considering that two out of the three species are sisters (occurring in geographical proximity). This close relationship raises concerns regarding the potential sharing of ancestral genetic variation or even introgression between these species that could explain the "convergent" patterns.

Minor points:

L81. See the comment above regarding convergence driving phenotypes and not the other way around.

L85. Clarify what you mean by "structural" here.

L93. and L106. The introduction could be improved by smoother transitions between paragraphs. New topics are introduced without any connection with the previous paragraph.

L111. How are rates of speciation/diversification relevant for the topic of the paragraph?

L117. We can detect higher rates of nonsynonymous substitutions in small populations but those are very often genetic load caused by genetic drift (i.e., deleterious alleles that can get fixed). High rates of nonsynonymous substitutions are not equal to higher rates of positive selection or adaptation.

L125. This is the first time in the introduction that the CT-scan data is introduced. It is unclear from the paragraph why this is being collected (e.g., the motivation).

L133. This is the first time it is mentioned in the introduction that certain species swallow fish whole. Which ones? Why is this interesting for the whole picture of the study?

L143. The results section mentions PSMC, but I could not find a segment in the methods describing this analysis.

L147. What exactly is the metric "shape of the curve" mentioned here?

L154. Again, the Boa paper does not show that islands "result" in genomic convergence. It shows that convergent traits in snakes that occur on island happens to be driven by convergent genomic changes and suggests that selective pressures in islands could play a role considering that the phenotypes of interest have clear connections to island adaptation.

L158. This is the prediction from neutral theory that should underlie the study hypotheses.

L186. Make sure the gene symbols are italicized throughout the text.

L226. See comment above regarding effective population size and natural selection according to neutral theory.

L275. Instead of mentioning RERconverge here, I suggest explaining what analyses the software performs.

L297. Please clarify what you mean by whole genome because you are only looking at coding sequence, which makes up ~1% of the genome.

L315. "Biological noise in the relationship between behavior and molecular convergence" sounds a little odd. Instead, it may be more appropriate to say that behavioral traits tend to be more polygenic, meaning they are influenced by multiple genes with smaller individual effect sizes. Consequently, the likelihood of convergent evolution in behavioral traits is reduced, as there are numerous alternative pathways for acquiring the same phenotype. It is worth noting that complex morphological traits can also be polygenic, but they may exhibit higher heritability compared to behavioral traits.

L334. I would be cautious when saying that there is more or less support for regulatory vs coding convergence based on only a handful of references.

L363. Same issue raised before here. It is crucial to clarify that population size alone does not directly drive positive selection. Instead, population size can influence the efficiency of selection. It would be more accurate to state that small population sizes are *associated* with higher levels of positive selection.

L417. How are the terms parallel and convergent different in the context of the study?

L433. "scaled by dividing by" sounds a little awkward. Maybe just scaled by $1/\pi^2/3$?

L437. Suggestion here: "We set up the cutoff of the posterior probability that a given residue is evolving convergently in plunge-diving lineages to 0.8"

L465. Did you select only genes whose w values were > 1 (some genes might be significant, but not under positive selection). I made that mistake before, so just make sure you double-check.

L500. Please clarify whether positive selection was determined for each species branch individually.

L511. Gene ontology enrichment analysis relies heavily on using an appropriate genomic background that accurately reflects the species annotation of interest, otherwise results can be extremely biased. Which background did you use? If not your own reference species, I would recommend finding a tool that takes custom background annotation or alternatively using PANNZER2 (Törönen et al 2018) to get gene ontology terms directly from proteins and then running a Fisher exact test manually for each GO term.

L520. It is not clear how many replicates were used in the morphometric analysis.

Figures 2 and 4. These are very nice figures. Well done!

Figure S9. It looks like there is a lot of phylogenetic signal on the number of positively selected genes. Many non-plunge-diving species show a high number of positively selected genes. Are there alternative explanations for this pattern?

Reviewer #4 (Remarks to the Author):

The paper sets out to examine convergent genomic evolution in plunge-diving kingfishers. The sample set was comprised of 31 species – nine of which were plunge-divers including three plunge-divers that occurred on islands. The remaining species were divided into island (six species) and non-island ones (11 species). The question is an interesting one, and the approach goes some way to help to understand the evolution of the feeding behaviour. However I have some major concerns and a suite of more minor ones. First, the paper over-reaches in places. For example, the small number of plunge-diving kingfisher species on islands means that any results that pertain to those specifically should be treated with a great deal of caution. Yet the main result the authors focus on (as per the title) is that there are stronger genomic signatures associated with a feeding style on islands. Second, the presentation and order of information is not always logical and easy to follow (expanded on below). I would like to have seen the focus the paper to be on the most well-supported analyses rather than over-reach on interpretations where the support is not as clear.

In the introduction, three aims are set out – to examine temporal patterns of population demography from whole genome sequences; to examine genomic convergence of the plunge-diving phenotype; and to examine relationships between tongue morphology and taste receptor genes. The background information that is used to set these aims up is highly variable and the logic not entirely clear. For example, for the second aim, a prediction is made that essentially says that because there are higher rates of non-synonymous mutations in island populations generally, that island kingfishers should experience stronger selection for plunge diving than continental ones. The necessary steps to link mutation rates and strength of selection are not made. Additionally, the efficiency of selection is theoretically predicted to be higher in large than small populations so it is not clear why the predictions about selection on islands are made. Further examples of where the paper structure could be improved are that for the first aim, the prediction is not set out until the results section (line 160), and the rationale for the third aim is a single sentence about penguins. Overall, the background material required for a reader to be fully informed about the three aims is not there.

The first result is that based on whole genome sequencing and demographic modelling, that island kingfishers have greater demographic fluctuations than mainland ones. Some red flags are raised in looking at figure S5. This shows an increasing N_e back through time on islands and a decrease back through time for continental forms, with the effective population sizes (assuming log to base 10) being very, very large, even for most recent estimates for islands. What are these estimates really capturing? It seems highly unlikely that island N_e sizes could have rivalled the sizes of those on

continents (in the many millions) and maybe the timeframe is extending back beyond the time of the island population existing in its insular version (note the x-axis label needs fixing). The authors conclude that these fluctuations have affected molecular evolution, but specifically how so is not made clear. The second main result is that there is genome-wide convergence in plunge divers (both continental and island), but not among all island forms. There are a suite of genes that associate with plunge diving that the authors make reasonable arguments to link to function in kingfishers. These are the strongest results and should be the main focus. Related to this is the finding positive genomic selection on the island plunge-divers is strong and shows more enriched gene functions. I am not convinced that this conclusion is very strong based on figures 4 and 5. As mentioned above, with only three island plunge-diving species, any result would need to be interpreted with extreme caution. Also those three appear to be part of a clade that includes continental plunge-divers, that have generally enriched gene function. To base the title of the paper on this finding given these considerations seems tenuous. Finally, there is a specific section about tongue morphology and taste receptor genes, which seems like a bit of an add-on (it was not introduced effectively).

Some minor comments:

Line 73 – This sentence is difficult to follow – is the species recognition trait also under divergent selection and/or responsible for dispersal?

Line 77 statement about plastic behaviours requires referencing

Line 106 - needs referencing

Line 107 – You could argue that they have a decreased variety of ecological interactions e.g. predator-prey, because islands tend to have fewer species, but this is not the same as fewer total ecological interactions.

Line 109 The sentence about genetic drift needs to be more precise. As genetic drift is stochastic, the implied influence is that it would reduce likelihood of genomic convergence.

Fig 3 – why were the sequences aligned to a carmine bee-eater instead of the new annotated *T. chloris* genome mentioned in the results?

Discussion – the first ~10 lines would be better placed in the introduction

The discussion contains lots of 'results' speak, including references to figures and tables.

Dear Dr. Eliason,

Your manuscript entitled "Genomic signatures of convergent shifts in feeding behavior are stronger on islands" has now been seen by 4 referees, whose comments are appended below. You will see from their comments copied below that while they find your work of potential interest, they have raised quite substantial concerns that must be addressed. In light of these comments, we cannot accept the manuscript for publication, but would be interested in considering a revised version that addresses these serious concerns.

We hope you will find the referees' comments useful as you decide how to proceed. Should further experimental data or analysis allow you to address these criticisms, we would be happy to look at a substantially revised manuscript. However, please bear in mind that we will be reluctant to approach the referees again in the absence of major revisions.

In particular, we ask that you:

- (1) Address all discussion points raised by the reviewers, while taking care to proofread and reformat the manuscript for clarity. At the same time, it would be important to qualify your conclusions, as mentioned by Reviewer #3-4 (acknowledging the limited number of island-dwelling plunge-divers, etc.).
- (2) Address the statistical concerns raised by Reviewers #1, 3-4, while also incorporating the additional descriptive figures requested by Reviewer #2.
- (3) We agree with Reviewers #3-4 that the data regarding taste receptor gene evolution and tongue morphology are not very well integrated into the text. We would recommend removing these results from the manuscript, or at least de-emphasizing these data (in part, by moving them to the Supplementary Information).

OUR RESPONSE: We have removed the tongue shape results from the manuscript.

We are committed to providing a fair and constructive peer-review process. Do not hesitate to contact us if you wish to discuss the revision or if there are specific requests from the reviewers that you believe are technically impossible or unlikely to yield a meaningful outcome.

If you decide to submit a revised version, we ask that you ensure your manuscript complies with our editorial policies. Please see our revision checklist for guidance on formatting the manuscript and complying with our policies. A comprehensive guide to our formatting requirements for final submissions is also available for your reference here.

Please use the following link to submit your revised manuscript, point-by-point response to the referees' comments (which should be in a separate document to any cover letter) and any completed checklist:

<https://mts-commsbio.nature.com/cgi-bin/main.plex?el=A3Cx4Gfs3A7BSas2I4A9ftdkA5ieZjaR4KSvXsrwQQs5AZ>

We expect major revisions of this nature to take around six months to complete, but appreciate that every situation is unique. Please take as long as necessary to address these concerns in full, including performing any additional experimental work required. We look forward to receiving your revised manuscript when it is ready and will not enforce any specific deadline. However, please bear in mind that if the revision process takes significantly longer than six months, we will need to confirm that nothing similar has been accepted for publication at Communications Biology or published elsewhere in the meantime.

Please do not hesitate to contact me if you have any questions or would like to discuss the required revisions further. Thank you for the opportunity to review your work.

Best regards,

George Inglis

George Inglis, PhD
Senior Editor
Communications Biology
orcid.org/0000-0002-9069-5242

Referee expertise:

Referees #1-4 all have expertise in avian evolution and genomics

Reviewers' comments:

Reviewer #1 (Remarks to the Author):

In this paper, the authors use a large set of 31 whole genomes of different species of kingfishers that independently evolved plunge-diving behaviors on islands or continental locations (representing populations of these species with different types of demographic history). The authors found that different types of kingfishers showed evidence of different overall demographic histories on islands and continents, and correspondingly, island plunge-divers had stronger signatures of positive selection than continental species. They found several different types of genes associated with convergence in plunge-diving birds, and that there is the rapid evolution of taste receptors, but no evidence that these shifts are associated with tongue shape.

Major Concerns:

Overall the authors did a thorough job of analyses, and use robust combinations of methods that support their findings. I also applaud them on a streamlined manuscript that is clearly written. My biggest concerns with this paper are not about the methods that they used, but about having had enough detail on the methodology to be able to figure out in detail what they did. Although I appreciate that it is the format of the paper, it was difficult to read the results section alone and understand what was going on without going back and reading the corresponding methods section. So, overall, I would recommend that the authors present a bit more of the methods when presenting the results. There was one section that especially needed more information on the methodology in the paper overall to be able to understand, which I go into further below:

OUR RESPONSE: We appreciate this point and have added several details throughout to allow the reader to replicate our results (see specifics below).

Methods Questions in the “Effective population size fluctuates more on islands” section:

- Was the topology of the phylogeny of these species estimated separately and held constant for the M0 model? If so, how was the species tree of those relationships estimated?

OUR RESPONSE: Yes it was, and we now state this in line 374. The species tree was estimated with UCE data using Bayesian and maximum likelihood approaches, as discussed in McCullough et al. (2019).

- The comparative analyses of the PSMC results were compelling, but I didn’t totally understand how they were conducted with the methodological detail that was provided. For example, how did they score the shape of the curve? For the number of population fluctuations, how was that determined? When comparing curve shape, did they consider uncertainty?

OUR RESPONSE: We added an additional paragraph in lines 322-336 describing the settings we used for PSMC and our approach of comparing curve shapes.

- Did the authors estimate effective population sizes of mainland versus island taxa, or was that inferred from the PSMC results? PSMC is not a good program for estimating modern population sizes, so they should be calculated separately (they could compare levels of heterozygosity among island and non-island taxa). Looking at Figure 1, it appears that in several cases there are big jumps to higher N_e , for example, in *Actenoides lindsayi*.

OUR RESPONSE: Given this issue with inaccurate N_e estimates in modern populations using PSMC, we wanted to focus on the overall shape of the PSMC curves rather than absolute N_e estimates. We now clarify this in lines 133-137.

We appreciate the suggestion of comparing heterozygosity levels, but we are not sure it would get at the hypothesis about Ne fluctuations that we're interested in testing. As such, we would prefer to leave this analysis as-is.

Minor Concerns:

Introduction, 4th paragraph: I would recommend reordering this paragraph, so the impacts of being on an island on patterns of genetic diversity and genomic convergence are closer to that introduction, before jumping in on speciation rates. Similarly, in the last sentence of that paragraph, do you mean genes related to plunge diving?

OUR RESPONSE: We have reordered the paragraph to put the components about convergence on islands first. We also clarified that we are talking about genes related to plunge-diving behavior in line 106.

Results, "Several genes are adaptively convergent in plunge-divers" section: How many genes were tested, and how many genes were identified as being under positive selection? I also think that the second paragraph describing specific functions would be more suited to the discussion versus the Results section.

OUR RESPONSE: We added a note about how many genes were tested (line 174) as well as moved the second paragraph to the discussion (lines 207-228).

Results, "Positive selection associated with plunge-diving is stronger on islands" – When talking about how island plunge-divers cluster together, is that relative to all species, or to other plunge-diverse? I think it would help the reader by mentioning that the tree shown isn't representative of genetic relationships, but rather similar go-terms.

OUR RESPONSE: Good idea. We added a line to the figure caption to this effect (lines 521-523).

Results, "Convergence in tongue shape is largely decoupled from taste receptor evolution": The reader could benefit from a brief description of the hyoid bone and how it might relate to important tongue morphology.

OUR RESPONSE: We have now removed this section, as suggested by the AE and other reviewers.

Figure 2 – the tree diagram showing convergent versus non-convergent pairs is very helpful, but it would be helpful to also define them briefly with words.

OUR RESPONSE: We added further details about the exemplar species pairs (see Fig. 2 caption lines 492-495).

Reviewer #2 (Remarks to the Author):

The manuscript entitled “Genomic signatures of convergent shifts in feeding behavior are stronger on islands” by Eliason et al. explored patterns of convergent evolution associated with the feeding behavior of 31 species of kingfishers. The authors report significant levels of convergent evolution, especially for species with plunge-dive behavior. The authors also report that island species had more dynamic effective population sizes through time and maintained smaller populations, which can directly impact patterns of positive selection. The genes under selection in the targeted species were associated with GO terms potentially important for the evolution of their behaviors. In addition to the genome-wide scans, the author found signals of positive selection in genes associated with taste receptors linked to fish-eating behavior in other species of birds. The manuscript is quite interesting, given that it explores the genomic architecture behind distinct feeding behaviors, which is not that common in the literature. Although the associations they found are not proof of the function of the genes, the authors have paved the way for future functional studies targeting specific genes and pathways. The manuscript is well-written, and the main ideas being tested are clear and easy to understand. Most of my comments are minor. For instance, I missed more detailed explanations on particular subjects that might help readers understand the main components of the paper (see comments below). The methods used in the manuscript are adequate, however, I was not able to check the reproducibility of the study given that the Dryad link with the scripts was not available.

OUR RESPONSE: We double-checked and ensured that all scripts and data sets are now available for review. We have the genomes uploaded to Zenodo in a private repository that will go live upon acceptance of the manuscript. The genome files are rather large (>2 GB) but we are happy to upload a single genome for review, if the reviewer would like.

I found the reduced association between CNEs and targeted traits quite interesting. On the evolutionary scale of the study, regulatory elements should play an important role in determining targeted traits. I wonder if there is an alternative way to map CNEs to the species' genomes other than using blast hits on the chicken genome. Maybe select CNEs between species that are more closely related to the targeted group?

OUR RESPONSE: We are not aware of any other available CNE data sets. We confirmed this with the lead of the Vertebrate Genomes Project (Erich Jarvis, pers. comm.). Our mapping approach using LAST recovered hits to >88% of those in chicken. These were likely not false positives as the LAST aligner is well-suited for aligning divergent species. In future work we are hoping to do a multi-species kingfisher genome alignment and identify CNEs specific to this group using cactus.

Minor comments:

L109: Explain how it might influence convergence. Lower N_e makes selection less effective, and we might expect reduced convergent patterns.

OUR RESPONSE: We reworded this paragraph to reflect this idea (see lines 97, 117-118).

L116: I think it is important to mention nearly neutral theory (Ohta 1973) here.

OUR RESPONSE: Done (see line 100).

L122: I think it is important to mention here the other variables associated with island species that can impact evolutionary rates (e.g., smaller sizes).

OUR RESPONSE: Good suggestion, it has been incorporated (see lines 107-109).

L139: It would be good to describe in the text (not only in Fig. 1) how many instances of plunge-diving behavior evolved in the system and how many times it occurred on islands vs. continents.

OUR RESPONSE: We have added this important information in lines 131-132.

L158: I think it is essential to include here the fact that drift is stronger in smaller populations, reducing the effectiveness of selection.

OUR RESPONSE: We have included this (e.g., line 150).

L163: An interesting piece of information that could easily be added to the paper is the location of genes associated with convergent evolution. Are there clusters of genes, or are they widespread across the genome? I wonder if genes with convergent amino acid substitutions in plunge-diving species are “linked” or not.

OUR RESPONSE: Great point. We had provided a Manhattan plot (Fig. 3C) to show the genomic distribution of positively selected genes, but we have now added a line describing that the pattern is roughly uniform throughout the genome (see line 176).

L226: How does this relate to nearly neutral theory and mutation load?

OUR RESPONSE: We modified this text to represent the idea that positive selection on islands might be more indicative of shifts in the direction of selection (enough to overcome drift from small pop sizes, see lines 194-198).

L399: Can you provide a graph showing the relationship between % of mapped reads and divergence time from the reference?

OUR RESPONSE: We have added this as new Figure S3.

Figure S3: I cannot read this figure. The information is too small.

OUR RESPONSE: Since the exact genomic coordinates are not important (but rather overall trends in depth relative to where positively selected genes are), we opted to remove them, make the tick marks wider, and add a description to the caption. We also removed the coverage depth values as they were redundant for each concentric circle.

Reviewer #3 (Remarks to the Author):

The article "Genomic signatures of convergent shifts in feeding behavior are stronger on islands" by Eliason et al. investigates signatures of genomic and morphological convergence in plunge-dive behavior across kingfishers and attempts to correlate these signatures with biogeographic characteristics of the species, specifically differentiating between continental and island-dwelling species. While the study offers an intriguing and relevant question, a robust dataset comprising 31 kingfisher genomes accompanied by morphometric measurements, and thorough analyses, there are certain concerns regarding the manuscript's focus, narrative, and hypotheses that need to be addressed. In the following sections, I will provide a detailed account of these concerns.

One of the primary weaknesses of the manuscript lies in its lack of focus. Although the authors have conducted a series of intriguing analyses, such as identifying amino acid changes occurring convergently across plunge-diver species, detecting signatures of convergent positive selection, testing the frequency of these genetic convergences in islands versus continents, examining signatures of positive selection in taste receptor genes, and exploring the correlation between tongue shape and evolutionary rates in taste receptor genes, these analyses seem disjointed and lacking a cohesive narrative. The introduction, in particular, fails to effectively establish how these questions will be addressed later in the text, giving the impression that the various analyses were just randomly assembled into a single manuscript. The introduction would benefit tremendously from a major rewriting that not only establishes the research questions but also provides a stronger background to justify the hypotheses being tested. For example, why would living on islands and having lower effective population size influence the likelihood of observing genomic convergence? Why does taste receptor gene evolution and its correlation with tongue morphology matter for the big picture of the article?

OUR RESPONSE: We have removed the tongue morphology component from the manuscript and reorganized/rewrote the introduction.

Another major concern pertains to the notion that species living on islands show smaller effective population sizes and therefore more positive selection and overall genomic convergence. My understanding is that, according to neutral theory, a lower effective population size implies that natural selection is less effective due to the predominant influence of genetic drift. Genetic drift can lead to the removal of adaptive variants or an increase in the frequency of deleterious alleles. This concept is distinct from the strength of selection, which is determined by environmental pressures (either biotic or abiotic) and can, in fact, be stronger on islands compared to continents (due to, for instance, intense competition for limited resources or other island's ecological dynamics).

OUR RESPONSE: Good point, we have clarified that with small N_e genetic drift should lower genomic convergence (line 97) but divergent selection among islands (or between continents and islands) could drive positive selection in genes important for island colonization, competition, etc. (e.g., lines 107-109, 196-198, 273-278).

Based on these considerations, it is reasonable to expect that a lower long-term effective population size on islands would result in fewer signatures of selection, unless the strength of selection is exceptionally high.

OUR RESPONSE: We agree and have now clarified (and corrected) this throughout the manuscript.

On top of that, I am struggling to understand the rationale behind the idea that genomic convergence should occur more often on islands. While it may be logical to expect increased convergence if island species exhibit a higher propensity for hybridization or if they share a common ancestor, I can't see why being on an island could influence the chances of evolving a trait through the same genetic change. The article would improve substantially if these points were clarified.

OUR RESPONSE: We have carefully read through all instances relating to our predictions of the amount and strength of molecular convergence/positive selection on islands and made several corrections throughout (e.g., line 96, 118).

My third concern is with the recurrent statement suggesting that a behavior or trait can "cause" molecular convergence (see L81, L154, L414). I think the reasoning here is completely inverted. Certain traits and behaviors can evolve due to convergent molecular changes and not the other way around. Please revise the text to accurately reflect the relationship between molecular convergence and the emergence of traits or behaviors, emphasizing that convergent molecular changes are the underlying drivers that enable the evolution of similar traits or behaviors across different species.

OUR RESPONSE: We have changed instances where our meaning was unclear and/or the logic was inverted (e.g., lines 144-147).

Another major point concerns the analysis of convergence associated with plunge-diving behavior. As suggested in Foote et al. (2015), cited in the manuscript, both neutral and adaptive substitutions occurring in genes are limited because of constraints related to deleterious or pleiotropic effects of these mutations. Consequently, coding sequence convergence can occur even in the absence of apparent shared positive selection. However, methods such as codeml in PAML, which are commonly employed to detect rates of change, are unable to test the null hypothesis that the foreground species (in this case, the plunge-diving species) have no relationship with the observed convergence in coding sequence. To adequately test the hypothesis of convergence, a recommended approach would be to employ a permutation test

where the foreground species would be randomized, and the analysis would be repeated multiple times. Then you could use the distribution of the number of convergent changes to infer whether the empirical result falls beyond what is expected by chance.

OUR RESPONSE: We appreciate this concern.

We would like to point out that we had discussed this issue with PAML in the original manuscript (lines 471-481) and added additional analyses that do not have this problem of foreground/background conflation as does PAML (e.g., Hyphy RELAX). We feel this approach is conservative, and we would therefore like to retain these original analyses in the main text.

In addition, we have added two new analyses of convergent positive selection. First, we added a “drop test” analysis recommended by Kowalczyk et al. (biorxiv) in which we removed foreground species (i.e., plunge-divers) and tested whether a M2 model of positive selection across the non-foreground species is supported. If it is, the branch-site test is considered not significant, but if the M2 model is non-significant this suggests that the pattern of positive selection is unique to the foreground species. Second, we used a more recent approach estimating both convergence and positive selection among branch pairs at the same time (CSUBST). The results of both of these analyses are now incorporated into the supplementary materials (e.g., see Supp. Methods para. 4, Results para. 4, Figures S7, S9, and Supp. data sets S3 and S4).

Finally, I would like to raise the issue concerning the small sample size of plunge-diving species occurring on islands, which consists of only three species. Beyond the statistical limitation, I have reservations about the robustness of the results, particularly considering that two out of the three species are sisters (occurring in geographical proximity). This close relationship raises concerns regarding the potential sharing of ancestral genetic variation or even introgression between these species that could explain the “convergent” patterns.

OUR RESPONSE: Although we are aware of other studies that have looked at shifts in suites of gene on single branches (e.g., Hahn 2015), we agree this can be problematic with only two origins of plunge-diving on islands. As such, we have refocused the manuscript primarily on the convergent evolution of genes in plunge-diving species and removed the phrase about islands from the title (“Genomic signatures of convergent shifts to plunge-diving behavior in birds”).

Minor points:

L81. See the comment above regarding convergence driving phenotypes and not the other way around.

OUR RESPONSE: Noted, and we have changed these occurrences throughout.

L85. Clarify what you mean by “structural” here.

OUR RESPONSE: This line has been removed.

L93. and L106. The introduction could be improved by smoother transitions between paragraphs. New topics are introduced without any connection with the previous paragraph.

OUR RESPONSE: We reorganized the introduction to improve readability and allow for more logical connections between paragraphs/ideas.

L111. How are rates of speciation/diversification relevant for the topic of the paragraph?

OUR RESPONSE: We removed this aspect from the introduction.

L117. We can detect higher rates of nonsynonymous substitutions in small populations but those are very often genetic load caused by genetic drift (i.e., deleterious alleles that can get fixed). High rates of nonsynonymous substitutions are not equal to higher rates of positive selection or adaptation.

OUR RESPONSE: We appreciate this point. We thought about our predictions, made several revisions to this section, and added additional references backing up the idea that natural selection is less efficient in small/island populations ((e.g., see line 103, refs. 36, 37).

L125. This is the first time in the introduction that the CT-scan data is introduced. It is unclear from the paragraph why this is being collected (e.g., the motivation).

OUR RESPONSE: We cut this component from the manuscript.

L133. This is the first time it is mentioned in the introduction that certain species swallow fish whole. Which ones? Why is this interesting for the whole picture of the study?

OUR RESPONSE: This has been cut in the revised version.

L143. The results section mentions PSMC, but I could not find a segment in the methods describing this analysis.

OUR RESPONSE: We have added a new paragraph (see lines 322-336).

L147. What exactly is the metric “shape of the curve” mentioned here?

OUR RESPONSE: We added further details about how we analyzed the shape of curves in the methods (line 330-331).

L154. Again, the Boa paper does not show that islands “result” in genomic convergence. It shows that convergent traits in snakes that occur on island happens to be driven by convergent genomic changes and suggests that selective pressures in islands could play a role considering that the phenotypes of interest have clear connections to island adaptation.

OUR RESPONSE: We appreciate this distinction and have revised the text as such (e.g., lines 144-147).

L158. This is the prediction from neutral theory that should underlie the study hypotheses.

OUR RESPONSE: Agreed, we have made several edits throughout the manuscript to make this evident that island species show show fewer convergent genomic shifts (e.g., line 97).

L186. Make sure the gene symbols are italicized throughout the text.

OUR RESPONSE: Done.

L226. See comment above regarding effective population size and natural selection according to neutral theory.

OUR RESPONSE: Please see our response above.

L275. Instead of mentioning RERconverge here, I suggest explaining what analyses the software performs.

OUR RESPONSE: This paragraph has been removed in the revised version.

L297. Please clarify what you mean by whole genome because you are only looking at coding sequence, which makes up ~1% of the genome.

OUR RESPONSE: We appreciate this point, and we have changed the text to “kingfishers are convergent genome-wide” accordingly (line 200).

L315. “Biological noise in the relationship between behavior and molecular convergence” sounds a little odd. Instead, it may be more appropriate to say that behavioral traits tend to be more polygenic, meaning they are influenced by multiple genes with smaller individual effect sizes. Consequently, the likelihood of convergent evolution in behavioral traits is reduced, as there are numerous alternative pathways for acquiring the same phenotype. It is worth noting that complex morphological traits can also be polygenic, but they may exhibit higher heritability compared to behavioral traits.

OUR RESPONSE: We appreciate the suggested wording and have incorporated it (see lines 241-244).

L334. I would be cautious when saying that there is more or less support for regulatory vs coding convergence based on only a handful of references.

OUR RESPONSE: We removed this line.

L363. Same issue raised before here. It is crucial to clarify that population size alone does not directly drive positive selection. Instead, population size can influence the efficiency of selection. It would be more accurate to state that small population sizes are *associated* with higher levels of positive selection.

OUR RESPONSE: We changed the text as suggested (e.g., see lines 271-278).

L417. How are the terms parallel and convergent different in the context of the study?

OUR RESPONSE: We are only interested in convergence for this study, and we now fixed the text to reflect that (line 340).

L433. “scaled by diving by” sounds a little awkward. Maybe just scaled by $1/\pi^{2/3}$?

OUR RESPONSE: We implemented the suggested wording in line 358.

L437. Suggestion here: “We set up the cutoff of the posterior probability that a given residue is evolving convergently in plunge-diving lineages to 0.8”

OUR RESPONSE: Thank you, we implemented the suggestion.

L465. Did you select only genes whose w values were > 1 (some genes might be significant, but not under positive selection). I made that mistake before, so just make sure you double-check.

OUR RESPONSE: We went back and double checked that the selected genes had $w > 1$.

L500. Please clarify whether positive selection was determined for each species branch individually.

OUR RESPONSE: Yes, we clarify this in line 423.

L511. Gene ontology enrichment analysis relies heavily on using an appropriate genomic background that accurately reflects the species annotation of interest, otherwise results can be extremely biased. Which background did you use? If not your own reference species, I would recommend finding a tool that takes custom background annotation or alternatively using PANNZER2 (Törönen et al 2018) to get gene ontology terms directly from proteins and then running a Fisher exact test manually for each GO term.

OUR RESPONSE: We now use the collared kingfisher background (see new Figures 3 and 4) and state that in the Methods (line 439).

L520. It is not clear how many replicates were used in the morphometric analysis.

OUR RESPONSE: This section has now been removed, following the AE's suggestion.

Figures 2 and 4. These are very nice figures. Well done!

OUR RESPONSE: We appreciate the kind remark! We put a lot of thought into the color scale to maximize interpretability among the figures.

Figure S9. It looks like there is a lot of phylogenetic signal on the number of positively selected genes. Many non-plunge-diving species show a high number of positively selected genes. Are there alternative explanations for this pattern?

OUR RESPONSE: Our PGLS analyses are designed to account for this phylogenetic signal. We already include body size, foraging behavior, and insularity as potential predictors of this trend—but if there are specific variables the reviewer has in mind we would be happy to consider adding them.

Reviewer #4 (Remarks to the Author):

The paper sets out to examine convergent genomic evolution in plunge-diving kingfishers. The sample set was comprised of 31 species – nine of which were plunge-divers including three plunge-divers that occurred on islands. The remaining species were divided into island (six species) and non-island ones (11 species). The question is an interesting one, and the approach goes some way to help to understand the evolution of the feeding behaviour. However I have some major concerns and a suite of more minor ones. First, the paper over-reaches in places. For example, the small number of plunge-diving kingfisher species on islands means that any results that pertain to those specifically should be treated with a great deal of caution. Yet the main result the authors focus on (as per the title) is that there are stronger genomic signatures associated with a feeding style on islands. Second, the presentation and order of information is not always logical and easy to follow (expanded on below). I would like to have seen the focus the paper to be on the most well-supported analyses rather than over-reach on interpretations where the support is not as clear.

In the introduction, three aims are set out – to examine temporal patterns of population demography from whole genome sequences; to examine genomic convergence of the plunge-diving phenotype; and to examine relationships between tongue morphology and taste receptor genes. The background information that is used to set these aims up is highly variable and the logic not entirely clear. For example, for the second aim, a prediction is made that essentially says that because there are higher rates of non-synonymous mutations in island populations generally, that island kingfishers should experience stronger selection for plunge diving than

continental ones. The necessary steps to link mutation rates and strength of selection are not made. Additionally, the efficiency of selection is theoretically predicted to be higher in large than small populations so it is not clear why the predictions about selection on islands are made

OUR RESPONSE: We appreciate this comment and have made several changes to clarify our predictions on islands (e.g., see lines 111-123).

Further examples of where the paper structure could be improved are that for the first aim, the prediction is not set out until the results section (line 160), and the rationale for the third aim is a single sentence about penguins. Overall, the background material required for a reader to be fully informed about the three aims is not there.

OUR RESPONSE: We now state the prediction earlier in Introduction line 117.

The first result is that based on whole genome sequencing and demographic modelling, that island kingfishers have greater demographic fluctuations than mainland ones. Some red flags are raised in looking at figure S5. This shows an increasing N_e back through time on islands and a decrease back through time for continental forms, with the effective population sizes (assuming log to base 10) being very, very large, even for most recent estimates for islands. What are these estimates really capturing? It seems highly unlikely that island N_e sizes could have rivalled the sizes of those on continents (in the many millions) and maybe the timeframe is extending back beyond the time of the island population existing in its insular version (note the x-axis label needs fixing). The authors conclude that these fluctuations have affected molecular evolution, but specifically how so is not made clear.

OUR RESPONSE: We now point out that these N_e curves are highly sensitive to input values of generation times and mutation rates, add a citation to this effect (Nadachowska-Brzyska et al. 2022 MEE), and state that we are focusing more on the shapes of the curves through time than the absolute numbers. We also added several methodological details in lines 322-336 about the PSMC approach we took.

The second main result is that there is genome-wide convergence in plunge divers (both continental and island), but not among all island forms. There are a suite of genes that associate with plunge diving that the authors make reasonable arguments to link to function in kingfishers. These are the strongest results and should be the main focus.

OUR RESPONSE: We agree and have reframed the manuscript, as well as removed the phrasing about islands from the title.

Related to this is the finding positive genomic selection on the island plunge-divers is strong and shows more enriched gene functions. I am not convinced that this conclusion is very strong based on figures 4 and 5. As mentioned above, with only three island plunge-diving species, any result would need to be interpreted with extreme caution. Also those three appear to be part

of a clade that includes continental plunge-divers, that have generally enriched gene function. To base the title of the paper on this finding given these considerations seems tenuous.

OUR RESPONSE: See above.

Finally, there is a specific section about tongue morphology and taste receptor genes, which seems like a bit of an add-on (it was not introduced effectively).

OUR RESPONSE: This has now been cut from the manuscript, following this suggestion and that of the AE.

Some minor comments:

Line 73 – This sentence is difficult to follow – is the species recognition trait also under divergent selection and/or responsible for dispersal?

OUR RESPONSE: This line has been removed (see below).

Line 77 statement about plastic behaviours requires referencing

OUR RESPONSE: This line has been removed since we were asked by the AE and other reviewers to streamline the manuscript by removing the CT scan results.

Line 106 - needs referencing

OUR RESPONSE: We added a cite to Losos and Ricklefs (2009).

Line 107 – You could argue that they have a decreased variety of ecological interactions e.g. predator-prey, because islands tend to have fewer species, but this is not the same as fewer total ecological interactions.

OUR RESPONSE: We removed this line from the manuscript.

Line 109 The sentence about genetic drift needs to be more precise. As genetic drift is stochastic, the implied influence is that it would reduce likelihood of genomic convergence.

OUR RESPONSE: We corrected this line to state that drift should decrease genomic convergence (e.g., line 118).

Fig 3 – why were the sequences aligned to a carmine bee-eater instead of the new annotated *T. chloris* genome mentioned in the results?

OUR RESPONSE: This was done because the bee-eater has chromosome annotations but the *T. chloris* genome does not (just scaffolds). We clarified this in the caption (see line 512).

Discussion – the first ~10 lines would be better placed in the introduction
The discussion contains lots of ‘results’ speak, including references to figures and tables.

OUR RESPONSE: We moved these 10 lines to the introduction (e.g., see lines 65-74) and removed ‘results’ speak when possible.

REVIEWERS' COMMENTS:

Reviewer #1 (Remarks to the Author):

The authors have done a commendable job of addressing my previous concerns, as well as those from other reviewers. The manuscript is now better-focused and it is more obvious how the different questions fit together. I agree with the decision to remove the tongue morphology analyses, which should be able to stand on their own in a subsequent publication. I appreciate the newly rearranged Introduction section, and my only further suggestions are a couple of minor suggestions for further clarifying some sections within.

Lines 287-289: It isn't clear how this new sentence relates back to plunge-diving and convergence. This paragraph would be much stronger if it related kingfishers as a system back to plunge-diving or the broader questions about behavioral convergence. I'd have that last new sentence integrated into the following paragraph.

Lines 304-309: This statement feels premature, given that the next paragraph states the goals of the study. I would recommend moving these predictions to the following paragraph after the main goals for the study and the questions being asked have already been established.

Reviewer #3 (Remarks to the Author):

I commend the authors for the substantial improvement in their manuscript, especially the introduction and overall narrative. It reads much nicer now and I'm happy to recommend it for publication.

Minor comments

L178 - Make sure this gene list is capitalized.

L332 - "we first clipped the time range to that covered by all 31 species" - not sure if this is a typo but it is unclear what you mean by this.

Reviewer #1

Lines 287-289: It isn't clear how this new sentence relates back to plunge-diving and convergence. This paragraph would be much stronger if it related kingfishers as a system back to plunge-diving or the broader questions about behavioral convergence. I'd have that last new sentence integrated into the following paragraph.

Our response: L287: As suggested, we moved this line to the following paragraph (L77-80) and revised the paragraph to emphasize why kingfishers are an ideal system for studying our broader questions about how behavior influences traits and genomes (e.g., see L68-75).

Lines 304-309: This statement feels premature, given that the next paragraph states the goals of the study. I would recommend moving these predictions to the following paragraph after the main goals for the study and the questions being asked have already been established.

Our response: We moved these lines to the following paragraph, as suggested (see L104-109).

Reviewer #3

L178 - Make sure this gene list is capitalized.

Our response: We assume the reviewer meant "italicized", and we have fixed this accordingly.

L332 - "we first clipped the time range to that covered by all 31 species" - not sure if this is a typo but it is unclear what you mean by this.

Our response: We changed the text to read "first determined the time range with Ne data for all 31 species" (L322)